# Artificial Intelligence in Evaluation of Permanent Impairment: New Operational Frontiers

**DOI:** 10.3390/healthcare11141979

**Published:** 2023-07-08

**Authors:** Roberto Scendoni, Luca Tomassini, Mariano Cingolani, Andrea Perali, Sebastiano Pilati, Piergiorgio Fedeli

**Affiliations:** 1Department of Law, Institute of Legal Medicine, University of Macerata, 62100 Macerata, Italy; r.scendoni@unimc.it (R.S.); mariano.cingolani@unimc.it (M.C.); 2International School of Advanced Studies, University of Camerino, 62032 Camerino, Italy; 3Physics Unit, School of Pharmacy, University of Camerino, 62032 Camerino, Italy; andrea.perali@unicam.it; 4Physics Division, School of Science and Technology, University of Camerino, 62032 Camerino, Italy; sebastiano.pilati@unicam.it; 5School of Law, Legal Medicine, University of Camerino, 62032 Camerino, Italy; piergiorgio.fedeli@unicam.it

**Keywords:** artificial intelligence, permanent impairment, International Classification of Diseases (ICD), International Classification of Functioning (ICF), machine learning

## Abstract

Artificial intelligence (AI) and machine learning (ML) span multiple disciplines, including the medico-legal sciences, also with reference to the concept of disease and disability. In this context, the International Classification of Diseases, Injuries, and Causes of Death (ICD) is a standard for the classification of diseases and related problems developed by the World Health Organization (WHO), and it represents a valid tool for statistical and epidemiological studies. Indeed, the International Classification of Functioning, Disability, and Health (ICF) is outlined as a classification that aims to describe the state of health of people in relation to their existential spheres (social, family, work). This paper lays the foundations for proposing an operating model for the use of AI in the assessment of impairments with the aim of making the information system as homogeneous as possible, starting from the main coding systems of the reference pathologies and functional damages. Providing a scientific basis for the understanding and study of health, as well as establishing a common language for the assessment of disability in its various meanings through AI systems, will allow for the improvement and standardization of communication between the various expert users.

## 1. Introduction

The scientific discipline of machine learning (ML), an intersection of statistics and computer science, explores how machines can learn from data [1]. In automatic learning, algorithms are employed to make predictions and, at the same time, to “learn” in the absence of instructions from static programs (as in the case of traditional software) through the elaboration of very large datasets, following various types of paths [2,3]. The advent of technologies based on artificial intelligence (AI) holds promise for the development of applications useful in the most varied disciplines, in particular medicine, just because of the capacity of these algorithms to process enormous quantities of data with a combined mechanism of execution of operations/self-learning to support healthcare professionals in decision making, with the objective of reducing errors through the integration of artificial intelligence in clinical practice [4].

To achieve this goal, there has been a rapid spread and progression of machine learning in recent years in various spheres of medicine, among them oncology, cardiology, genomics, imaging diagnostics, forensic pathology, and integrated homecare services [5,6,7,8]. Some types of algorithms based on deep learning, such as convolutional neural networks, can be applied in imaging diagnoses of some tumors or can distinguish between malignant cutaneous neoplasms through the elaboration of clinical images of cutaneous neoplasm, showing high precision comparable or superior to that of expert humans [9,10]. 

In this context of great expectations and technological ferment, it is natural to consider the application of machine learning to evaluate impairment caused by a trauma or disease, given its importance in a wide variety of situations, such as compensation for injury caused by a wrongful act or compensation in the sphere of private or social insurance. Currently, the quantification of permanent psychological or physical impairment is assessed exclusively by the medico-legal expert, who evaluates the impairment based on a substantially subjective analysis of evidentiary and clinical data and formulates a conclusive impairment rating related to reference parameters in tables or guidelines that differ in various spheres and countries [11,12,13,14]. This evaluation is of fundamental importance because, albeit in different forms, it provides the basis for awarding benefits in the various spheres mentioned above. For that matter, even though the medico-legal expert is highly trained and uses specific tables, subjectivity can lead to inaccuracies in the evaluations, for example, errors due to incorrect analysis of the data available or the methodology followed in complex cases such as multiple disabling situations in the same subject [13,15]. Thus, it is to be augured that soon it will be possible to benefit from the support of AI in the evaluation of psychological–physical impairment, hopefully, to reduce the margin of error in evaluation by “educating” the algorithms through the use of big quantities of data. Some countries have recognized this need and are in the early stages of efforts to optimize disability evaluation and citizen services. Of note in this context is the French government’s attempt to develop a database to serve in the development of an algorithm for personal injury compensation [16,17]. 

This work lays the groundwork for proposing an operational model for the use of AI in evaluating impairments with the ultimate goal of making the data processing system as uniform as possible, starting from the principal systems of coding for pathologies and functional damage with reference, respectively, to the International Classification of Diseases (ICD) and the International Classification of Functioning, Disability, and Health (ICF) [18,19,20]. These classifications, already widely used for statistical and planning purposes regarding public health spending, are also used as an instrument for evaluating disability. Thus, this project of AI-assisted disability evaluation will be based on the ICD and ICF classifications.

The first objective of the work is to standardize the indications coming from the ICD and ICF using the respective alphanumeric codes to uniquely identify the diagnostic expressions (ICD codes disorders) and correlated life situations (ICF codes) also through the use of the scale qualifiers envisaged by the ICF. 

This preliminary result of data standardization will be followed by the use of a set of concrete cases through which to apply a supervised deep learning path. At the end of the training, the system will therefore be provided with a set of “test” cases to evaluate its efficiency. At the end of the process, the system will subsequently be used with the automatic production of a quantitative assessment of the damage in each case.

This first contribution aims to address the first part of the project, laying the foundations for choosing the most suitable artificial intelligence tool and testing it preliminarily on a few indicative cases to verify the stability of the data and their usability for deep learning.

## 2. Overview of AI Systems in Biomedical and Medico-Legal Fields

The medico-legal application of machine learning currently has relatively few literature items when compared to other medical fields. This is especially true considering that a large portion of the research focuses on forensic pathology and forensic sciences, such as forensic toxicology, forensic genetics, forensic odontology, and so on. A systematic review conducted by Galante et al. (2023) identified 72 relevant medico-legal articles, including 12 on forensic odontology, 19 on forensic pathology, 21 on forensic genetics, and only 5 on other branches, including ethical issues [20].

Research in the field of AI and legal medicine is predominantly directed towards issues related to forensic sciences, with little to no interest in impairment assessment (and, for example, medical liability), which constitutes a significant part of medico-legal practice in some countries.

However, it should be noted that a literature search on the application of AI in disability assessment, using the PubMed platform and the search string “(disability) AND (assessment) AND (artificial intelligence),” yielded 654 results. Nonetheless, only one study addresses the possibility of utilizing artificial intelligence for disability evaluation by an assessment committee. This study, conducted by Vasudeva et al. (2021), proposes AI as a fairer solution for disability assessment by medical commissions through the integration of telemedicine and artificial intelligence with the International Classification of Functioning, Disability, and Health (ICF) [18]. The aim is to make the evaluation process more transparent.

The presence of robotic systems for the assistance and rehabilitation of individuals with spinal cord injuries is particularly noteworthy. Several systematic reviews have already shown the potential of these systems to improve walking speed, distance covered, muscle strength, range of motion, and mobility in patients with spinal damage [21].

There are also studies that theorize the evaluation of motor impairment in diseases such as Parkinson’s for purposes other than compensation. For instance, the KELVIN system has shown results comparable to those of expert evaluators, although its use is mainly clinical [22].

However, it is important to consider that the current research on the application of artificial intelligence in disability assessment mainly focuses on the theoretical aspects of rehabilitation rather than purely evaluative medico-legal purposes. Nevertheless, these studies still involve the application of validated scales as the basis for the programming and functioning of the machine [23,24,25].

Artificial neural networks (ANNs), including, e.g., multi-layer perceptrons (MLPs), convolutional neural networks (CNNs), Bayesian convolutional neural networks (BCNNs), and general adversarial networks (GANs), are computational analytical tools inspired by the biological nervous system. Arguably, they represent state-of-the-art models in the field of machine learning, but they typically require large datasets for training. They consist of interconnected computing units, in jargon, called “neurons”, that perform parallel computations for data processing and knowledge representation. CNNs, in particular, are specifically designed for image processing tasks. In fact, CNNs with three-dimensional filters can also be used for video analysis. In general, ANNs are commonly trained via the backpropagation algorithm, whereby differences between actual and predicted outputs are backpropagated to optimize the network’s weights and biases. BCNNs employ Bayesian inference for probability computations [20]. The most standard use of ANNs is performing classification and regression tasks. Notably, generative ANNs can be implemented too. Typical examples are so-called restricted Boltzmann machines, variational autoencoders, or large language models. These networks can learn the probability distribution of training examples and generate additional realistic examples from the estimated distribution.

In Figure 1, a schematic representation is presented depicting the functioning of an artificial neural network (ANN) applied to the assessment of disability percentage based on medical examination, ICD code, ICF code, and evaluation Barémes; note the three layers of information integration.

Beyond ANNs, many other more conventional statistical models are often used in data science for classification, regression, or clustering. Popular choices are, e.g., decision trees (DTs), random forests (RFs), support vector machines (SVMs), and k-nearest neighbors (k-NNs). DTs are tree structures where internal nodes represent attribute tests, branches represent test outcomes, and leaf nodes hold class labels. RFs combine the outputs of multiple decision trees to reach a final result. SVMs use a training set to find a hyperplane that produces the largest minimum distance (margin) between objects of different classes. k-NNs predict values based on the similarity of features between new data points and points in the training set [20]. 

Several research articles have already discussed the adoption of machine learning tools in legal medicine. Quite naturally, many of these studies employed conventional statistical models. For example, Yang et al. (2020) used support vector machines (SVMs) to classify skulls based on morphological features, while random forest (RF) models were employed by Navega et al. (2015) for ancestry estimation [26,27]. Such methods are suitable when the available training datasets are not large, say, of the order of tens and up to a few thousand training instances. Notably, also more ambitious deep learning algorithms, i.e., those based on deep ANNs, have been adopted. Indeed, deep neural networks have often proven capable of extracting useful information from massive datasets, but they might be affected by overfitting phenomena when the available training datasets are sparse. For example, some authors employed CNNs to perform age estimation from pelvic radiographs. As mentioned above, convolutional models are particularly suitable for scanning images or three-dimensional representations [28]. To partially cope with the size of the training dataset (only of the order of thousand instances), the parameters of a pre-trained network (the AlexNet model) were imported, and only the last fully connected layers were re-trained. This strategy is, by now, quite common in the field of image analysis, and it is often referred to as transfer learning. Popular CNNs (GoogLeNet, DANet, and DASNet) have also been used by Bewes et al. (2019) to analyze dimorphic skull features and by Vila-Blanco et al. (2020) for age estimation [29,30]. 

It is worth stressing that this brief overview of selected articles is not meant to be exhaustive. Applications of machine learning techniques in the broader field of medicine are rapidly growing, in particular since the spread of the COVID-19 pandemic [31,32,33,34]. An emerging problem, beyond the limited size of the training datasets, is the occurrence of unidentified statistical correlations between training and testing data, possibly combined with other anthropogenic factors in the data collection [35]. These effects can lead to overestimated performances of the trained statistical models, which are not replicated when the test is performed in more realistic scenarios [26,36]. A promising coping strategy is represented by synthetic data produced via generative neural networks [37]. While these also stand for promising strategies to address privacy problems, they unavoidably involve possible side effects, as the generated datasets might, in turn, be affected by biased models [38,39]. 

## 3. The ICD as a Possible Base for AI

The International Statistical Classification of Diseases, Injuries, and Causes of Death (ICD), developed in the last century by the World Health Organization (WHO), is a standard of classification of diseases and related problems. Since alphanumeric codes indicate the medical terms used to formulate disease diagnoses and diagnostic procedures and therapies, the ICD is the main instrument used today for comparative statistical and epidemiological studies on the causes of morbidity and mortality among places and over time. The most recent version, ICD-11, was adopted by the World Health Assembly in 2019 and took effect on 1 January 2022 [19]. The ICD forms the foundation of healthcare statistics and is used in many spheres, including administration, healthcare research, and the planning of healthcare assistance. The ICD provides semantic interoperability and enables the re-utilization of data not only for simple healthcare statistics but also for decision-making support, resource allocation, reimbursement, guidelines, and other operational spheres. However, there are some limitations to this classification system for codifying a set of clinical information in an ICD code. First, some terms are generic and fail to capture the substantial differences between similar pathologies. Second, the system is too dependent on the operator for the assignment of codes, and there can be inaccuracies in matching the pathologies and the codes [39].

In order to overcome the inaccuracies intrinsic to the ICD system, new deep learning models have been proposed, based on the possibility of automatically translating medical diagnoses into the corresponding codes through, for example, recurrent neural networks that can distinguish automatically between the different types of ICD and capture hidden semantic information precisely, ensuring precise automatic ICD codification based on the same healthcare documentation [39,40]. 

Given its universality and succinctness, the ICD could serve as the first step in defining a classification based on the sequence of etiology, pathology, and clinical manifestation, in order to evaluate disability based on artificial intelligence systems. This is even more relevant in the current scenario where research seeks to use deep learning neural networks to limit the error intrinsic in the passage from medical diagnosis to ICD classification. Thus, for the construction of a process for precise disability estimation, the ICD appears to be eligible as the first step in the codification of healthcare information, followed by integration with other classification systems to contextualize the diagnosis, with reference to the ICF (the International Classification of Functioning, Disability, and Health), already recommended by the WHO to provide an interactive and multidimensional estimate of invalidity [41]. 

## 4. Second Degree of Evaluation of Disability and the ICF Classification

The International Classification of Functioning, Disability, and Health, also created by the WHO, more commonly known as the ICF, is a classification of health and health-related domains. Since an individual’s functioning and disability should be contextualized in a specific social setting, the ICF also contains a list of environmental factors. The ICF is based on the same principles as the ICD and thus shares the same set of extension codes, making it possible to document pathologies in greater detail [42].

In fact, the diagnosis of a pathology alone cannot define the patient’s disabilities, the level of needed assistance, or the functional outcomes. The existence of the disease does not in and of itself foretell the person’s occupational performance or the potential for and the probabilities of occupational and social reintegration. Instead, the ICF synthesizes medical and social factors and thus provides a summary of the biological, individual, and social aspects of the disease [42]. Specifically, the ICF coding uses an alphanumerical system in which the various components, grouped into chapters, are indicated by the letters “b” (body) for body functions, “s” (structure) for body structures, “e” (environment) for environmental factors, and “d” (domain) for activities carried out and participation. These letters are followed by a numeric code that begins with the number of the chapter (a figure), followed by the second level (two figures), and by the third and fourth levels (a figure each) [41]. 

See Figure 2. which provides a schematic representation of the steps involved in assigning the ICF code, specifically focusing on an example of impairment in mental functions.

Then the ICF categories are inserted so that the broadest categories include more detailed subcategories. This structure ensures that each individual can have a series of codes at each level that can be independent or correlated. The goal of this system is to establish a classification of human functioning and disabilities [43]. The ICF classification adopts a biopsychosocial paradigm of disability in alternative to both medical and social ones; the fact that it is multiform makes it a better tool for the interpretation of disabilities. The ICF, created by the WHO, is the current framework of reference for the description of disabilities as it links the state of disease and impairment, or more simply, health and function [44,45]. 

For some time now, some countries have been examining the possibility of using the ICF to define criteria for the personalization of personal injury so that this may contribute to formulating technically motivated evaluations [8,46,47,48]. When an evaluation of personal injury is requested for purposes of compensation, for example, in cases of car accident injuries, the amount of the compensation, at least in some countries, is based on the evaluation of the impairment and its negative impact on the dynamic-relational sphere. In this context, the ICF proposed by the WHO could prove to be a useful instrument for combining impairment with environmental factors. The use of the ICF in the near future can be greatly improved with the implementation of new technologies like AI to bridge gaps and reduce contrasts among physicians, legal professionals, and insurance companies [18,49].

Body functions: assessing the specific impairments or dysfunctions in physiological and psychological functions that contribute to the disability.Activity limitations: evaluating the difficulties an individual may face in executing tasks or actions due to the disability.Participation restrictions: examining the limitations in an individual’s involvement in life situations and social interactions resulting from the disability.Environmental factors: considering the external factors, such as physical, social, and attitudinal, that can either facilitate or hinder an individual’s functioning and participation.

Note: The ICF (International Classification of Functioning, Disability, and Health) provides a comprehensive framework for classifying disabilities. Code b11420 corresponds to a specific classification based on the interrelated levels of body functions, activity limitations, participation restrictions, and environmental factors. Each level offers valuable insights into understanding and addressing the impact of disabilities on an individual’s functioning and engagement in various aspects of life.

## 5. Impairment Evaluation and Prospects for Using Machine Learning

There are many medical and legal operative spheres that require an evaluation in deciding whether to grant a benefit. Narrowing the discussion to Italy, these decisions regard compensation for personal injury due to wrongful acts, compensation in the sphere of private insurance and the sphere of government worker’s compensation program (INAIL, the National Institute for Insurance against Accidents at Work), and the concession of benefits in the social welfare sphere. In all these operative spheres, when an individual presents a psychological–medical impairment, it is necessary to carry out an evaluation to determine the degree of disability. This activity is always conducted by evaluation experts through the use of a table of reference.

Currently, impairment evaluation methods are based on the medico-legal expert’s subjective interpretation of the clinical and instrumental data. This interpretation is then related, according to the specific sphere in which the evaluation is requested, to the indications in the specifically developed guidelines or tables in order to determine an impairment rate. This quantification of the disability rate changes according to the sphere of evaluation and to the nation where it is conducted (different countries have different rules for the evaluation of disability, according to the local law). A common concern is the possibility of error due to the inevitable subjectivity of human opinion in every measurement of disability [18]. This concern significantly predates the development of the new technologies of AI and machine learning, not only in the evaluation of disability related to compensation but also in the evaluation of impairment for purposes of rehabilitation. As far back as 1933, in their praiseworthy work, Lim et al. postulated the use of AI, then in its infancy, through the development of a traditional type of software able to evaluate damage to the limbs on the basis of data acquired during the objective examination based on the *Guides to the Evaluation of Permanent Impairment, Third Edition* of the American Medical Association (AMA) [38]. While futuristic for its time, it nonetheless presented somewhat frequent evaluation errors due to the tables used as a reference and to the computers, which lacked systems of learning characteristic of modern machine learning. Now we find ourselves in the final phase of the itinerary of medical-legal evaluation, in which it may be possible to use AI to further decrease the risk of errors. Once correct ICD and ICF classifications have been input, the situation of disability thus defined can be related to the specific context in which the evaluation is requested. 

Currently, as mentioned above, tables and guidelines remain the most used method for calculating the percentage of an individual’s disability. In general, this system assigns a value to each anatomic region and type of dysfunction based on the degree of impairment. In general, there are also indications for combining a number of values in the case of impairments to multiple apparatuses [50,51]. Thus, the guidelines and tables constitute the instrument for converting medical information on disabilities into numerical values. They have the advantage of providing a standardized mechanism for the quantification of a given physical or mental condition, assuming that the evaluation is conducted by recognized experts. However, notwithstanding the usefulness of guidelines and tables, criticisms have not been lacking, among them the difficulty of always providing a complete, valid, and entirely trustworthy system. In addition, at times, the tables provide elements that are insufficient for defining basic components for the evaluation of the consequences of the impairment in terms of other dysfunctions [52,53,54]. 

In fact, today, all the guidelines and tables for the evaluation of psychological and physical impairment offer a reference tool for physicians who must conduct an evaluation. The possible use of new AI technologies does not aim to do completely away with table systems but to use their information as a point of departure for building an AI system structured on elements widely shared by the scientific community, able to provide greater objectivity, uniformity, and precision in the conclusive evaluation requested. It is to be hoped that the advent of machine learning can limit possible errors and overcome the defects of the current evaluation system, and eventually even allow automatic input of the evaluation of domains (activity, participation) and environmental factors, as amply mentioned in the international classification of functional and health disabilities. 

## 6. Operational Aspects and Hypotheses

The functional distinction between traditional data elaboration and automatic learning is that in the latter, a model learns from concrete examples rather than operating through foundational rules, an itinerary useful in medicine [55]. Thus, using algorithms for learning from observations or examples, the computers determine how to carry out a mapping from features to labels in order to create a general model that makes it possible to correctly carry out an activity with new input not seen before. According to the functioning of the programmed learning, subcategories of learning can be postulated, namely the principal ones of supervised learning and unsupervised learning, as well as reinforced learning and semi-supervised learning [56].

The objective of supervised learning is to predict a known *output* or *target*, for example, the recognition of handwritten figures or the classification of images of objects. These are tasks that a trained person can perform well and in which the machine approximates human performance. In other words, supervised learning concentrates often on classification, which entails the choice among subgroups to best describe a data string, and on regression, through the estimation of an unknown parameter [1]. Instead, in unsupervised learning, there are no instances already associated with the correct label or corresponding target value. Rather, unsupervised learning seeks to find archetypal models or natural groupings in the data. This activity is intrinsically more complex to evaluate, and often the validity of these groupings is judged only a posteriori, responding to the question of whether the model predicted by the machine is somehow useful [57]. Irrespective of the chosen method, the use of AI to analyze great quantities of data enables the creation of a new system to be constantly trained, with the possibility of obtaining ever more precise results to supplement or substitute human evaluations. 

The present work proposes new instruments for the evaluation of personal injuries, with the implementation of machine learning in the medico-legal field and the creation of specific algorithms to estimate the extent of personal injuries based on the usual evaluation tables, widely used nosological and functional classification systems (ICD and ICF), as well as the broadest possible databases about ratings of personal injuries from public institutes or private entities, or other sources. 

Figure 3 illustrates a schematic representation of the evaluation process proposed to estimate the percentage of damage. The example presented demonstrates the evaluation process for a dorsal–lumbar spine fracture.

Consequently, a possible evaluation of impairment using machine learning must be structured into phases. For example, the development of a preliminary function of impairment estimation seems to be more functional in the case of permanent rather than temporary impairment. Again, the construction of an algorithm for the estimation of a single anatomic region seems easier to carry out ab initio than a complex multi-regional evaluation. 

In fact, currently, there are no standard models for impairment estimation by AI, and thus, at least in the beginning, it is necessary to obtain a standardized and versatile algorithm that can be widely used in many instances of evaluation. Clearly, the first goal is to obtain a calculation system endowed with the capacity for “active learning,” unlike some calculation programs based on “static” software for personal injury assessment, which carry out stereotyped calculations. As in standard supervised learning campaigns, the dataset has to be structured in the following format: ${\left\{x_{i},y_{i}\right\}}_{i=1}^{N}$. In this formula, the vector $x_{i}$ includes the features, selected from the ICD and/or ICF classification systems, that describe each impairment instance; each vector is associated with the corresponding value of the impairment quantification $y_{i}$, which is, at least in a first step, provided by the medical expert; finally, the integer index i=1,…,N labels the N instances in the database. 

Having concluded this first phase of data gathering and manipulation in the multidisciplinary sphere of legal medicine, computer science, and statistics and having input the main table reference systems, the developers should proceed to the next phase of the study, exploiting the (hopefully large) number of already concluded personal injury evaluations to start the actual process of learning by the system based on real experience. This should take place in two steps, first using data obtained from traditional evaluations and, if successful, moving on to having the machine carry out the same evaluations in a system of “positive reinforcement” of abilities founded on the same executed performances. In the first step, the goal of the training of the machine learning model is to learn the mapping function $f\left(x\right)=y$ so that, after training, the impairment quantification for a previously unseen impairment/injury instance can be predicted via the function $f\left(x\right)$. Suitable candidate machine learning models are RF and SVM regressors. It is worth mentioning that these models are available from the versatile and popular machine learning library named Scikit-learn, which represents the ideal software for the early development stage [58]. The optimization of the model parameters can be performed by minimizing the mean-squared error on a suitably sized training set. Special care has to be devoted to identifying possible (in fact probable) overfitting phenomena, whereby the model just memorizes the training data but fails to accurately generalize to previously unseen instances. Following standard practices, the whole dataset shall be split into training, validation, and test sets, which are used for model training, hyper-parameter selection (e.g., number of trees in RF), and performance evaluation, respectively. Standard metrics, such as the coefficient of determination $R^{2}$, can be adopted to quantify the performance of the test set. In a subsequent development stage, when the number of training instances increases, more sophisticated regression models could be adopted; chiefly, we point to the use of dense neural networks (i.e., multi-layer perceptrons). In the final (and more ambitious) step, reinforcement or self-supervised learning protocols could be adopted, exploiting generative neural networks, such as GANs. Also, semi-supervised training can be envisioned, whereby expert users further tune the (pre-trained) network, giving a grade to the network predictions, as performed nowadays in the development of large language models such as the well-known GTP networks. Thus, the general objective is to create a system that can be widely used by medico-legal experts in routine processes of impairment evaluation in order to verify its practical applicability and then compare the estimates obtained by artificial intelligence and those by “conventional” evaluations.

### Ethics

The implementation of new AI-powered systems in permanent impairment assessment necessitates exploring the ethical issues accompanying this imminent paradigm shift.

The use of artificial intelligence (AI) in assessing personal damage raises important ethical questions. While the use of AI models can offer significant advantages in terms of efficiency and standardization of assessment, it is crucial to carefully consider the potential risks and implications arising from this practice [59].

Firstly, AI in personal damage assessment can lead to depersonalization of the decision-making process. The absence of direct human interaction may reduce the individual undergoing assessment to objective and numerical data, disregarding the complexity and uniqueness of the person. It is important to ensure that the use of AI does not result in dehumanizing treatment towards the individuals involved and does not disregard the subjective suffering of the person [60,61].

Secondly, the issue of transparency and interpretability of AI models emerges as an ethical concern. Since machine learning algorithms often operate in complex and non-linear ways, understanding how decisions are made and which factors influence those decisions can be challenging. This raises concerns about accountability and the possibility of challenging assessments made by AI. It is necessary to ensure that algorithms are developed transparently and that the individuals involved have access to clear and understandable explanations of the decisions made. This becomes particularly relevant, especially in legal contexts where a specific outcome may be contested by one or more parties involved.

Another ethical aspect concerns the potential presence of biases and discrimination within AI models. If the data used to train such models are incomplete or biased, AI may perpetuate and amplify these biases, leading to injustices in assessing personal damage. Ensuring balanced and representative data collection, as well as careful validation of AI models, is essential to avoid systemic discrimination [20].

There is the issue of autonomy and human control. While AI can provide valuable support in personal damage assessment, it is important that the final decision remains in the hands of the expert evaluator. AI should be used as a complementary tool to assist experts rather than replace them.

Finally, it is imperative to ensure that AI systems respect the privacy of individuals being assessed and maintain data security. This involves taking measures for data protection, anonymization or pseudonymization of personal information, consent to data processing already employed by some private insurance institutions, and individuals’ control over their own data. Furthermore, it is important to establish clear rules and regulations for the responsible use of data in the context of AI in order to balance technological innovation with the protection of individual privacy.

## 7. Conclusions

This work seeks to verify the concrete feasibility of programming a personal injury evaluation algorithm that is concretely usable in the practice of medico-legal experts, or in other words, to set up a computer base containing the most common systems of classification of diseases and disabilities, together with the main evaluation criteria currently in use. Such a system could then be “trained” with information from databanks of private and/or government institutions in order to conduct a preliminary verification of the functionality of the method in performing at least basic impairment evaluations and, in parallel, to carry out the function of automatic learning in machine learning technology. 

## Figures and Tables

**Figure 1 healthcare-11-01979-f001:**
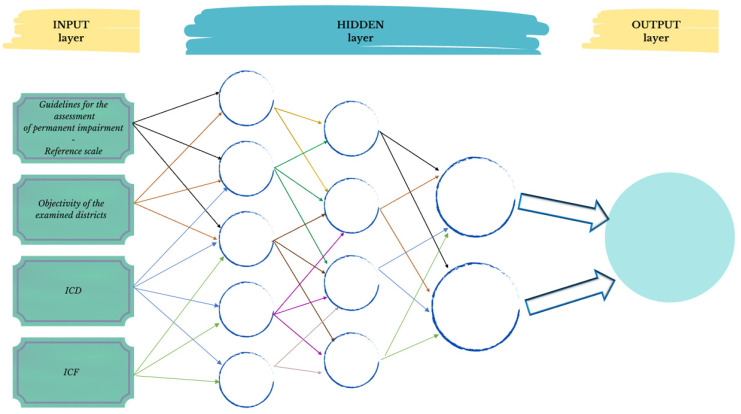
Hypothesis of ANN for estimating permanent impairment: bridging objective findings, reference scales, and WHO’s ICD and ICF codes.

**Figure 2 healthcare-11-01979-f002:**
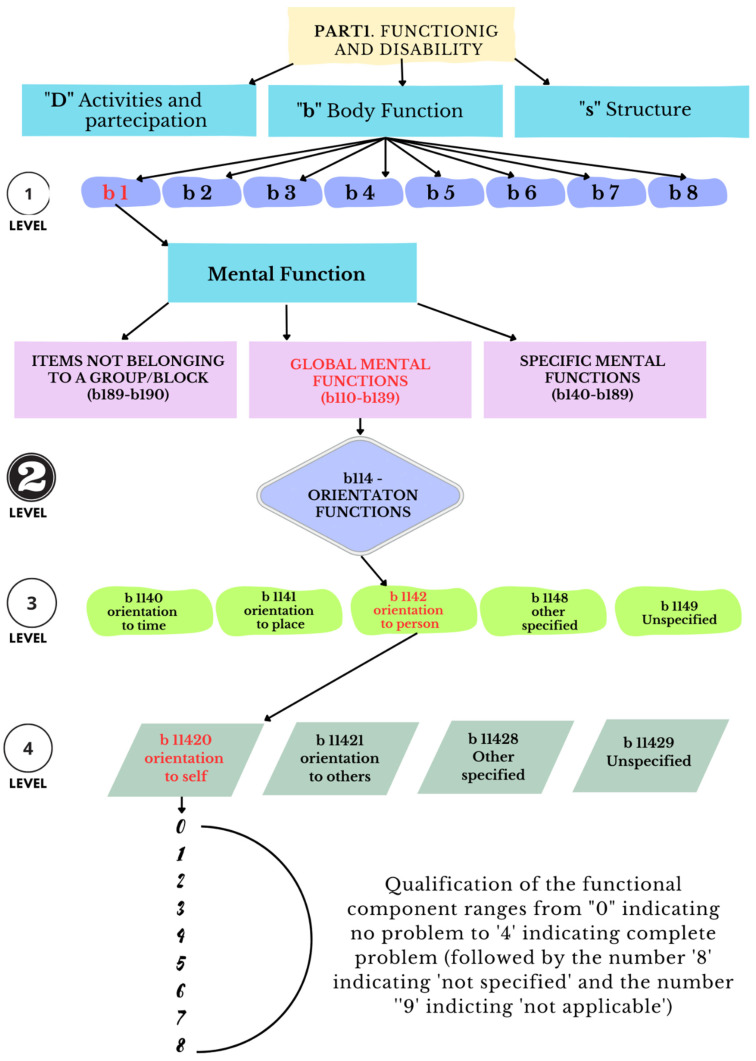
Illustration of the ICF functioning system for disability classification (code: b11420)—four levels of classification.

**Figure 3 healthcare-11-01979-f003:**
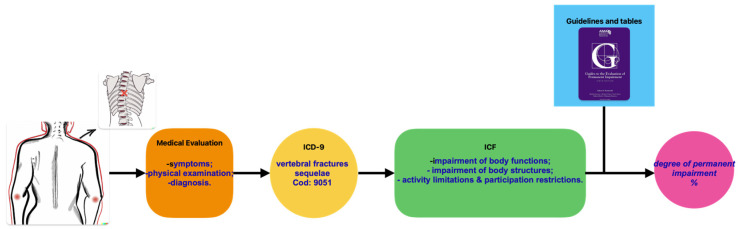
Schematic representation of the evaluation process proposed to estimate the percentage of damage.

## Data Availability

Not applicable.

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
