# Peer review of "Artificial Intelligence in Evaluation of Permanent Impairment: New Operational Frontiers"

_healthcare, 2023, doi:10.3390/healthcare11141979_

Round 1

Reviewer 1 Report

Artificial intelligence (AI) and machine learning (ML) span multiple disciplines, including the medico-legal sciences, also with reference to the concept of disease and disability. In this context, the International Classification of Diseases, Injuries and Causes of Death (ICD) is a standard for the classification of diseases and related problems, developed by the World Health Organization (WHO) and represents a valid tool for statistical studies and epidemiological. Indeed, the International Classification of Functioning, Disability and Health (ICF) is outlined as a classification that aims to describe the state of health of people in relation to their existential spheres (social, family, work).

The authors lay  the foundations for proposing an operating model for the use of AI in the assessment of impairments with the aim of making the information system as homogeneous as possible, starting from the main coding systems of the reference pathologies and functional damages.

The authors are convinced that providing a scientific basis for the understanding and study of health, as well as establishing a common language for the assessment of disability in its various meanings through AI systems will allow for the improvement and standardization of communication between the various expert users.

This is an interesting perspective.

I have some minor comments with a pure academic spirit.

1.      The aim is “Thus, this project of AI-assisted disability evaluation will be based on the ICD and ICF 78

classifications.” I suggest to better explain the purpose. Use bullet points to detail if you need

2.      Use the MDPI standard for the text. For example the introduction is fragments by a lot of white lines

3.      The citation of bibliography needs corrections. There are errors and/or an uncorrect use of the MDPI standards.  Use “[]”. See for example “Bewes et al. (2019) to analyse skull dimorphic features and in Vila- 120 Blanco et al (2020) for age estimation[27,28]. 121 It is worth stressing that this brief overview of selected articles is not meant to be exhaus- 122 tive. Applications of machine learning techniques in the broader field of medicine are rap- 123 idly growing [Baker2023]”

4.      Figure 1 and 2 need improvements and must be described in details in the text.

Author Response

Reply To Reviewers

Dear reviewers,

We would like to express our heartfelt gratitude to each of you for your valuable contributions in the review process of our article. We are deeply thankful for your dedication and the time you have invested in carefully examining our work.

Your suggestions and observations have played a crucial role in enhancing the quality of our paper. Thanks to your insightful evaluations, We have been able to make significant improvements and enrich the scientific content of the article.

The following changes have been approved based on your suggestions.

To Reviewer 1:

  • The aim is “Thus, this project of AI-assisted disability evaluation will be based on the ICD and ICF 78

classifications.” I suggest to better explain the purpose. Use bullet points to detail if you need.”

A part relating to more detailed purposes has been added in the introduction

  • Use the MDPI standard for the text. For example the introduction is fragments by a lot of white lines.

More text formatting has been done and much of the white lines have been eliminated.

  • The citation of bibliography needs corrections. There are errors and/or an uncorrect use of the MDPI standards.  Use “[]”. See for example “Bewes et al. (2019) to analyse skull dimorphic features and in Vila- 120 Blanco et al (2020) for age estimation[27,28]. 121 It is worth stressing that this brief overview of selected articles is not meant to be exhaus- 122 tive. Applications of machine learning techniques in the broader field of medicine are rap- 123 idly growing [Baker2023]”

The citations have been corrected and all put into the form "[]".

  • Figure 1 and 2 need improvements and must be described in details in the text.

The graphics of the figures have been improved and the description of each one has been inserted in the text. a third figure was also added.

Thank you again for giving us the opportunity to improve our manuscript.

Reviewer 2 Report

Overall, the paper addresses an important topic regarding the utilization of artificial intelligence (AI) in the evaluation of permanent impairments. The authors discuss the significance of the International Classification of Diseases (ICD) and the International Classification of Functioning, Disability, and Health (ICF) in this context. The paper provides a foundation for an operating model for the use of AI in impairment assessment and aims to enhance communication and standardization among expert users. While the paper has potential, there are several major revisions that need to be addressed before it can be considered for publication.

  1. Clarify the research objectives: The paper lacks a clear statement of research objectives. It is important to clearly state the specific goals of the study and what the authors aim to achieve through the proposed operating model for AI in impairment assessment.
  2. Provide a more comprehensive literature review: The authors briefly touch upon the role of AI and machine learning in the medico-legal sciences, but a more thorough literature review is necessary. The review should encompass recent studies, methodologies, and advancements in AI and ML applied to impairment evaluation. This will help contextualize the proposed operating model and highlight the novelty of the research.
  3. Expand on the methodology: The paper lacks a detailed description of the proposed operating model for AI in impairment assessment. It is crucial to provide a step-by-step explanation of how AI and machine learning techniques will be utilized in the evaluation process. Additionally, discuss the data sources, sample sizes, and any potential limitations or biases associated with the methodology.
  4. Provide examples and case studies: The inclusion of practical examples and case studies will strengthen the paper and demonstrate the effectiveness of the proposed operating model. Present real-world scenarios where AI can be applied to assess impairments, and discuss the outcomes and implications of using AI in those cases.
  5. Discuss ethical considerations: AI applications in the medico-legal field raise important ethical concerns. Address the potential ethical implications and challenges associated with the use of AI in impairment evaluation. Discuss issues such as data privacy, bias, transparency, and accountability, and propose strategies to mitigate these concerns.
  6. Conclusion and future directions: The paper should include a more comprehensive conclusion summarizing the key findings and contributions of the study. Additionally, provide a discussion on the future directions of research in this field, such as potential advancements in AI and machine learning techniques that could further enhance impairment evaluation.

Recommendations:

  1. Revise the introduction to clearly state the research objectives and the significance of the study.
  2. Conduct a more comprehensive literature review, incorporating recent studies and advancements in AI and ML applied to impairment evaluation.
  3. Expand on the methodology section, providing a detailed explanation of the proposed operating model and addressing any potential limitations or biases.
  4. Include practical examples and case studies to illustrate the application of AI in impairment assessment.
  5. Discuss the ethical considerations associated with AI in the medico-legal field and propose strategies to address them.
  6. Provide a comprehensive conclusion summarizing the key findings and contributions, and discuss future directions for research in this area.

Addressing these major revisions will greatly strengthen the paper and enhance its potential for publication.

English presentation should be improved.

Author Response

Dear reviewers,

We would like to express our heartfelt gratitude to each of you for your valuable contributions in the review process of our article. We are deeply thankful for your dedication and the time you have invested in carefully examining our work.

Your suggestions and observations have played a crucial role in enhancing the quality of our paper. Thanks to your insightful evaluations, We have been able to make significant improvements and enrich the scientific content of the article.

The following changes have been approved based on your suggestions.

  • Clarify the research objectives: The paper lacks a clear statement of research objectives. It is important to clearly state the specific goals of the study and what the authors aim to achieve through the proposed operating model for AI in impairment assessment.”

A part relating to more detailed purposes has been added in the introduction.

  • “Provide a more comprehensive literature review: The authors briefly touch upon the role of AI and machine learning in the medico-legal sciences, but a more thorough literature review is necessary. The review should encompass recent studies, methodologies, and advancements in AI and ML applied to impairment evaluation. This will help contextualize the proposed operating model and highlight the novelty of the research.”

A literature review containing the limited material related to AI in impairment assessment has been included in section 2, as requested.

  • “Expand on the methodology: The paper lacks a detailed description of the proposed operating model for AI in impairment assessment. It is crucial to provide a step-by-step explanation of how AI and machine learning techniques will be utilized in the evaluation process. Additionally, discuss the data sources, sample sizes, and any potential limitations or biases associated with the methodology.”

The request has been entered trying to satisfy your request.

  • “Provide examples and case studies: The inclusion of practical examples and case studies will strengthen the paper and demonstrate the effectiveness of the proposed operating model. Present real-world scenarios where AI can be applied to assess impairments, and discuss the outcomes and implications of using AI in those cases.”

what was asked was inserted in the text, as far as possible, and an image referring to concrete case was inserted (Fig.3).

  • “Discuss ethical considerations: AI applications in the medico-legal field raise important ethical concerns. Address the potential ethical implications and challenges associated with the use of AI in impairment evaluation. Discuss issues such as data privacy, bias, transparency, and accountability, and propose strategies to mitigate these concerns.”

A final sub-paragraph dealing with ethical considerations has been inserted.

  • “Conclusion and future directions: The paper should include a more comprehensive conclusion summarizing the key findings and contributions of the study. Additionally, provide a discussion on the future directions of research in this field, such as potential advancements in AI and machine learning techniques that could further enhance impairment evaluation.”
  • From all the changes inserted it is hoped that this aspect will be clearer

Thank you again for giving us the opportunity to improve our manuscript.

Round 2

Reviewer 2 Report

Dear Authors,

Dear Authors,

I would like to inform you that the review of your article has been completed, and I wanted to note that the revisions you made have brought the article to an acceptable level for acceptance.

Thank you, and best wishes for your continued success.

English should be checked for the last time.